# Clinical Effectiveness and Pharmacokinetics of Dalbavancin in Treatment-Experienced Patients with Skin, Osteoarticular, or Vascular Infections

**DOI:** 10.3390/pharmaceutics14091882

**Published:** 2022-09-06

**Authors:** Giacomo Stroffolini, Amedeo De Nicolò, Alberto Gaviraghi, Jacopo Mula, Giuseppe Cariti, Silvia Scabini, Alessandra Manca, Jessica Cusato, Silvia Corcione, Stefano Bonora, Giovanni Di Perri, Francesco Giuseppe De Rosa, Antonio D’Avolio

**Affiliations:** 1Infectious Diseases Unit, Department of Medical Sciences, University of Turin, 10149 Turin, Italy; 2Laboratory of Clinical Pharmacology and Pharmacogenetics, Department of Medical Sciences, University of Turin, 10149 Turin, Italy

**Keywords:** dalbavancin, long-acting, Gram-positive, PK/PD, osteoarticular infections

## Abstract

Dalbavancin (DBV) is a lipoglycopeptide approved for the treatment of Gram-positive infections of the skin and skin-associated structures (ABSSSIs). Currently, its off-label use at different dosages for other infections deserves attention. This work aimed to study the clinical effectiveness and tolerability of DBV in outpatients with ABSSSIs, osteoarticular (OA), or other infections, treated with either one or two 1500 mg doses of dalbavancin, for different scheduled periods. A liquid chromatography–tandem mass spectrometry method was used to measure total DBV concentrations. PK/PD parameters and the clinical and microbiological features of this cohort were evaluated in order to investigate the best predictors of treatment success in real-life settings. Of the 76 screened patients, 41 completed the PK study. Long-term PK was comparable to previous studies and showed significant differences between genders and dosing schedules. Few adverse events were observed, and treatment success was achieved in the vast majority of patients. Failure was associated with lower PK parameters, particularly C_max_. Concluding, we were able to describe DBV PK and predictors of treatment success in selected infections in this cohort, finding DBV C_max_ as a possible candidate for therapeutic drug-monitoring purposes, as well as highlighting the dual-dose one-week-apart treatment as the optimal choice for OA infections.

## 1. Introduction

Dalbavancin (DBV) is a semi-synthetic, novel, long-acting lipoglycopeptide active against Gram-positive pathogens, including multi-drug resistant isolates, approved for the treatment of ABSSSIs [1]. Long elimination half-life (around 9–10 days) and good tissue penetration represent the main pharmacokinetic features of dalbavancin, allowing for its long-term efficacy despite the simplified weekly administration regimens [2]. This extremely long half-life is also due to its high percentage of binding to plasma proteins (mainly albumin), which was described to vary from nearly 98% in rats to 93% in humans. DBV exhibits peculiar PK/PD properties, with good tissue penetration and a high susceptibility rate (near 100.0%) [2,3,4,5,6,7]. The clearance of DBV is not influenced by inhibitors or inducers of cytochrome P450, and therefore, its potential for drug–drug interaction involving hepatic metabolism is very low [2]. Furthermore, in an exploratory study, patient demographic characteristics had minor impacts on the pharmacokinetic profile of DBV, while females showed slightly higher concentration values, particularly in relation to body surface area (BSA) and body mass index (BMI) [3]. The ratio between the mean free area under the curve and the minimum inhibitory concentration (fAUC/MIC) was previously proposed as the best PK/PD parameter for its correlation with in vivo efficacy of DBV [2,3]. The 24 h fAUC/MIC values for net stasis, 1-log kill, and 2-log kill against *Staphylococcus aureus* were, respectively, 27.1, 53.3, and 111.1 [8,9]. No DBV dose adjustment is required in mild–moderate renal impairment, any degree of hepatic impairment, and for different modalities of renal replacement therapy (RRT); nevertheless, evidence in different modalities of RRT is poor, and the role of hypoalbuminemia still deserves better understanding [10], while dose reduction is suggested in patients affected by severe renal impairment. DBV exhibits a potent activity in vitro against the established biofilms due to *Staphylococcus aureus*, *Staphylococcus epidermidis*, and vancomycin-susceptible *Enterococci*, thus possibly playing a crucial role in the management of relevant infections characterised by bacterial biofilm production [11,12,13]. Nicolau et al. [14] reported a mean dalbavancin penetration into skin blister fluid of 59.6%, compared with plasma, after a single 1000 mg infusion, resulting in tissue concentrations well above the MIC_90_ of common Gram-positive pathogens implicated in ABSSSIs (including MRSA) for up to 7 days. In another study, Rappo et al. [15] found that, in 35 healthy subjects receiving a single infusion of 1500 mg DBV, the penetration into the epithelial-lining fluid was 36%, resulting in DBV lung concentrations exceeding the MIC90 of *Streptococcus pneumoniae* and *Staphylococcus aureus* for at least 7 days. Moreover, Dunne et al. [16] found a mean bone/plasma AUC of 13.1%, suggesting that two doses of DBV 1500 mg infusion administered one week apart may provide tissue exposure over the MIC for *Staphylococcus aureus* for 8 weeks. These findings were confirmed in two recent studies involving patients with OA infections [3,8]. Consequently, these convenient PK/PD properties make DBV a valuable alternative to daily in-hospital intravenous or daily continuous outpatient antimicrobial regimens in the treatment of long-term Gram-positive infections, posing the basis for its use beyond approved indications [4,5]. The DBV PK/PD relationship has been described to be AUC/MIC-dependent, making the application of TDM particularly difficult. In this scenario, some works suggested alternative analysis methods to the DBV quantification throughout treatment, such as the evaluation of the bactericidal activity of plasma samples [17], C_max_ values as a proxy of AUC [18], or punctual DBV concentration after 1 month [9]; recently, our group described DBV AUC over the long term in a small subset of patients with ABSSSIs or OA infections [3]. Despite all these studies, the definition of optimal PK/PD parameters with a useful TDM potential is still tricky, particularly considering patients with different clinical and microbiological features, as well as different dosing and posology. Therefore, more information is needed about DBV PK/PD and clinical effectiveness in a “real-life” context, in relation to the infection site and posological differences. The aim of this work was to better describe DBV PK in real-life settings and to characterise its PK/PD profile, for both approved and unapproved indications, treated with either a single 1500 mg dose or a double 1500 mg dose, 1 week, 2 weeks, or 3 weeks apart. Moreover, we aimed to evaluate the possible role of a TDM-based approach in optimising therapeutic choices, focusing on the most important PK/PD parameters for DBV, namely C_max_ and AUC/MIC, in different treatment periods. Ultimately, we aimed to link these PK/PD data to clinical outcomes and monitor the DBV safety profile.

## 2. Materials and Methods

### 2.1. Patient Enrolment and DBV Administration

We included patients with Gram-positive infections, with previous therapeutic failure to other antibiotic regimens or need for consolidation, and eligible for treatment with DBV, enrolled in the “Appropriatezza Farmacologica della Terapia Anti-Infettiva” (ethical approval no. 0040388 23 April 2020) clinical study. Patients stopped all their previous antibiotic therapies and were selected for a single 1500 mg or dual 1500 mg DBV infusion (1500 mg, 2 doses 1, 2, or 3 weeks apart) based on guidelines, investigator judgment, and the emerging data from the literature [19,20,21,22,23,24,25,26,27]. DBV was generally prescribed for patients who did not significantly benefit from previous therapies, for consolidation purposes, or due to intolerance to other therapies; therapy was conducted as an outpatient modality, also considering the COVID-19 emergency; a therapeutic scheme of two doses one week apart was recommended mainly in the setting of osteoarticular infections based on the data from other Italian and international experiences [3,8,19]. DBV preparation and administration were performed in this study as per label indications since patients’ selection and follow-up were in line with the instructions previously followed by our study group and described in an exploratory study [3]. Drug infusion was intravenously performed for 30 min. The cure was defined as the resolution of symptoms, microbiological cure, or no evidence of pathology as assessed by radiology techniques when applicable. Conversely, clinical/microbiological failure was retained if patients required additional antibiotics for their lack of response, presented new purulence, needed amputation, or died. The body surface area (BSA) was estimated using the Du Bois formula. Based on DBV administration, we identified 4 subgroups in relation to the interval that elapsed between the first and subsequent drug administration: *group 1* (1 week apart), *group 2* (2 weeks apart), and *group 3* (3 weeks apart); patients receiving a single DBV administration pertain to *group 0*.

### 2.2. PK and PD Evaluation

Blood sampling for the PK was performed in a 7 mL lithium/heparin tube using the following timing schedules, relatively to DBV infusion: 0 h (pre-dose), 0.5 h (end of infusion), 1 h, 1 week, 2 weeks, 3 weeks, 4 weeks, and then every 2 weeks, up to 6 months. No specific susceptibility testing was available for the direct determination of DBV MIC; therefore, surrogate categorisation of susceptibility was performed based on the MIC value for vancomycin, as previously reported and indicated by the EUCAST and Jones et al. [28]. On this basis, susceptibility to vancomycin (minimal inhibitory concentration, MIC 2 mg/L) was interpreted as a proxy of DBV MIC lower than 0.125 mg/L (nearly 97% probability from [28]), the EUCAST-suggested breakpoint for cocci. PK evaluation was performed as previously described in the literature [3,29], with an analytical kit (KIT-SYSTEM Antibiotics, CoQua Lab, Torino, Italy). Briefly, 50 µL of sample underwent protein precipitation, with the addition of isotope-labelled DBV as internal standard, and the diluted supernatants were analysed via reverse phase ultra-high performance liquid chromatography and tandem mass spectrometry (UHPLC–MS/MS). This method was associated with the mean bias and imprecision both being lower than 15%, in accordance with the EMA and FDA guidelines [29,30,31]. All the AUC data were calculated using a “linear-up/log-down” model by interpolating the concentration data of each patient, using the Phoenix WinNonlin software (Certara, Princeton, NJ, USA). The cumulative AUC_0–1w_/MIC, AUC_0–2w_/MIC, and AUC_0–4w_/MIC values were calculated in this analysis, in order to compare the different dosing schedules in terms of the overall early systemic exposure, which is supposed to be the most important for microbiological and clinical success [3,8,9]. Additionally, protein-binding-adjusted (PBA) AUC/MIC values were calculated by dividing the PK parameters by the PBA breakpoint MIC 1.786 mg/L (0.125 mg/L/0.07) considering a 93% protein binding, based on the mean percentage of protein binding in humans, as reported in several works [2,3,32].

### 2.3. Statistical Analysis

All statistical analyses were performed through Excel and SPSS 27.0 (IBM Corp, Armonk, NY, USA). Descriptive data are reported as percentages, median, and interquartile ranges (IQRs). The correlations between continuous data were evaluated through Pearson’s correlation tests. Differences between the groups were tested through the Kruskal–Wallis and Mann–Whitney non-parametric rank tests, and with an ANOVA test.

## 3. Results

### 3.1. Population Characteristics

A total of 76 patients were included in the preliminary screening and analysis. Of these, 41 completed the intensive PK study as per the Methods section (see Appendix A). The main clinical and demographic characteristics are presented in Table 1, and the main infection features are presented in Table 2. None of the involved patients was immunocompromised. Overall, chronic osteomyelitis was present in five patients. One patient dropped out because of an alternative diagnosis (tuberculous osteomyelitis). All the patients received previous antibiotic treatment except three (4%), comprising 52 patients (68%) with one single therapeutic line, 20 with two lines (26%), and 1 patient with three lines. The median duration of previous therapies was 28 days (IQR 14–56). Most commonly, patients were pre-treated with lipopeptides, glycopeptides, and beta-lactams. The reasons for treatment with DBV were: consolidation (43 patients, 56%), intolerance (2 patients, 3%), or failure (31, 41%) to other therapies. The majority of our patients (40, 54%) received two doses, while the remainder received one. The median time to cure after DBV was 56 days [28–84]. Overall, in terms of the final outcome, 58 patients (82%) were cured, 13 (17%) failed, and 4 (5%, excluded from the final analysis) were LFU patients, classified as per described in the Methods section. 

### 3.2. Pharmacokinetic Results

Of these 76 screened patients, 41 had a complete PK study, with a median of 7 [5–9.5] PK evaluations throughout the study. Of these patients, 25 (60%) were male; 17 patients (41.5%) were included in the one-dose dosing group (3 failed, 14 were cured); 8 patients (19.5%) were included in the two-dose, one-week-apart dosing group (*group 1*, no failure, 8 were cured); 13 patients (31.7%) were included in the two-dose, two-week-apart dosing group (*group 2*, 6 failed, 7 were cured); and 2 (4.9%) and 1 (2.4%) patients were included in the two-dose, three-week- and four-week-apart dosing groups (*group 3* and *4,* all cured). In addition, 20 patients (49%) underwent DBV therapy for consolidation purposes, 19 (46%) for failure, and 2 (5%) because of intolerance to other therapies. Eight patients from this group (20%) had diabetes, and three (7.3%) had slight renal insufficiency. Twenty-two (53.7%) underwent infection control procedures. Overall, 32 patients (78%) were cured at the end of the study period. The median PK parameters of the first dose, the cumulative AUC in the first month of therapy, as well as the protein-binding-adjusted (PBA) AUC/MIC (PBA AUC/MIC) results, categorised by dosing groups, are presented in Table 3 and displayed in Figure 1. Moreover, the curves describing the terminal half-life and mean PK parameters in each group are displayed in Appendix A. Second-dose AUC0–1w values were 32,810 mg/L·h [31,477–40,392]; 23,391 mg/L·h [21,930–30,780]; and 28,847 mg/L·h [28726–28847] for group 1, 2, or 3, respectively; second-dose AUC0–2w values were 45,595 mg/L·h [41,210–58,551]; 30,070 mg/L·h [29,183–40,644]; and 36,848 mg/L·h [36,636–36,848] for these same groups. Second-dose AUC0–3w values were 54,330 mg/L·h [46,317–68,054]; 40,570 mg/L·h [32,817–48,132]; and 41,124 mg/L·h [41,099–41,124] for the same groups. A significant positive correlation between Cmax at the first dose and cumulative AUC_0–4w_, for both *group 0* and *group 1* (*p* = 0.018; *p* = 0.041), was observed; C_max_ at the second dose for *group 1* also significantly correlated with the cumulative AUC_0–4w_ (*p* = 0.039). Moreover, C_max_ at the first dose in *group 1* was significantly correlated with AUC_0–1w_ (*p* = 0.015). The latter variable showed a significant positive correlation with cumulative AUC_0–4w_ for that group (*p* = 0.043).

A significant positive correlation between C_max_ at the first dose and cumulative AUC_0–4w_, for both *group 0* and *group 1* (*p* = 0.018; *p* = 0.041), was observed; C_max_ at the second dose for *group 1* also significantly correlated with cumulative AUC_0–4w_ (*p* = 0.039). Moreover, C_max_ at the first dose in *group 1* was significantly correlated with AUC_0–1w_ (*p* = 0.015). The latter variable showed a significant positive correlation with cumulative AUC_0–4w_ for that group (*p* = 0.043).

We performed the Kruskal–Wallis and Mann–Whitney non-parametric tests and found that the cumulative AUC_0–4w_ of treatment was significantly higher for *group 1*, compared with all the other dosing groups (*p* < 0.001; Figure 1). Moreover, at the second dose, *group 1* also showed a significantly higher C_max_ than *group 2* (*p* = 0.05; Table 3).

### 3.3. Association between DBV PK and Treatment Outcomes

Among all the PK parameters which were tested for their associations with treatment outcome, C_max_ at the first dose, AUC_0–1w_, and cumulative AUC_0–4w_ all resulted in higher values in cured patients, although these differences were not statistically significant (*p* = 0.114, 0.063 and 0.324, respectively; Figure 2). Concurrently, the highest efficacy was highlighted in *group 1*, especially when compared with the other groups, with a solid trend toward significance (*p* = 0.053, cured vs. not cured). The same PK parameters were tested for their associations with treatment outcome, further stratifying groups by infection site. When analysing ABSSSIs, no significant association was observed. For all the other indications, PK parameters appeared slightly reduced but without reaching statistical significance. This stratified analysis was not applicable to LVAD, because all three patients failed, nor did it apply to intravascular infections and spondylodiscitis, as all patients were cured at the end of the study period.

When stratifying this analysis by reasons for treatment with DBV (previous failure, intolerance, or consolidation) and site of infection, no significant difference was observed in patients with ABSSSIs, while for all the other indications, a significantly higher value for C_max_ after the first infusion was associated with treatment success for patients treated for consolidation purposes (*p* = 0.04) but not when patients were selected for therapy with DBV because of previous failure or intolerance (*p* = 0.214, Figure 3).

### 3.4. Correlations of DBV PK Parameters with Patient Characteristics

Overall, albumin was inversely correlated with the cumulative AUC_0–4w_ (*p* = 0.047). BSA was significantly and negatively correlated with AUC_0–1w_ at the first dose, C_max_ at the second dose, and the cumulative AUC_0–4w_, (*p* = 0.006, 0.002, and 0.004, respectively). C_max_ at the first dose showed only a trend towards a correlation with BSA (*p* = 0.091). No other significant correlations were found between eGFR, albumin, or other variables and DBV PK parameters. The Mann–Whitney test was applied relative to PK variables and gender in the whole cohort who underwent the PK study: Significant differences were found for C_max_ at the first dose and AUC_0–1w_ values with regard to gender, resulting in higher values for both variables in female patients (*p* < 0.001). Gender differences in PK parameters are depicted in Figure 4. These differences were clearly explainable by a difference in BSA, which was also significantly different between genders (*p* = 0.002). Nevertheless, these differences did not result in significant differences in outcome (*p* = 0.380), probably due to the small sample size and the low number of failures.

### 3.5. Predictors of Failure to DBV

In the overall population of 71 patients, in the context of off-label indications (OA and vascular infections), patients’ results were more likely to indicate failure to DBV if they were treated for failure to previous regimens (*p* = 0.036) and when treated for LVAD infections (*p* = 0.007). While no significant PK predictors of treatment success were observed in patients with failure to previous treatments, in the group of patients who were treated for consolidation purposes, failure was related to significantly lower DBV C_max_ at the first dose (*p* = 0.04), with a possible utility for future TDM purposes. For this reason, a ROC curve for C_max_ after the first dose was calculated with the aim to identify a threshold value capable to identify patients at risk for failure. From this analysis, the identified value was 313 mg/L, with the potential to identify patients prone to failure, with a sensitivity of 100% and a specificity of 78% (*p* = 0.035).

### 3.6. Testing Late PK Thresholds

In previous works, based on the PK/PD target attainment of an AUC/MIC ratio over 24 h after 1 month of treatment, two putative punctual threshold values were proposed for TDM purposes, 4 or 8 mg/L (in case of bacterial MIC of 0.062 or 0.125 mg/L, respectively) [8,9]. Considering these values, 7/41 and only 1/41 patients resulted below the 8 mg/L and 4 mg/L thresholds in our cohort, respectively. The latter patient experienced treatment failure. On the other hand, among patients with concentration values over the proposed 8 mg/L threshold, eight failed, while six patients with DBV concentrations below the 8 mg/L cut-off achieved treatment success. Finally, there was no statistically significant difference between DBV concentrations at week 4 with regard to outcome (*p* = 0.389). 

## 4. Discussion

In this work, we described the PK variability and clinical performance of DBV in its real-life use, taking into account its different posology for the treatment of OA infections, ABSSSIs, and intravascular/bloodstream infections. These data were analysed for associations and/or correlations with the demographical, anthropometrical, microbiological, and clinical features of each patient; more importantly, we analysed the relationship between DBV PK and therapeutic outcome. The results of the observed PK and clinical data were concordant with previous studies [3,8,27]. Our cohort is representative of a third-level referral hub since the enrolled patients had frequently a complicated clinical history related to the management of their infections, usually with failure to one or more therapeutic approaches (multi-failed/experienced patients). Additionally, a considerable number of microbiological isolates were resistant to common first-line agents (MRSA, MRSE). Considering this setting, our results are extremely encouraging, showing an 82% cure rate, confirming the high efficacy of DBV even for its off-label alternative indications. This success is remarkable also in view of the high prevalence of comorbidities, in particular diabetes (25%). The overall propensity to the adoption of source control/surgery (47% in this cohort) is noteworthy in these contexts, highlighting the importance of surgery in complicated infections and the advantages of a combined approach. Unfortunately, our centre lacks a joint collaboration with orthopaedicians, a factor that would possibly improve the infection control tendency and outcomes. Generally speaking, few patients were lost to follow-up, and time-to-cure for patients was in line with what is expected for the infections described in this study. Cumulative exposure to DBV in the first month of treatment (the most critical period to achieve bacterial eradication at the site of infection) revealed different results in patients in different exposure groups (close versus delayed re-administration). PK variables seemed generally higher in cured patients, although the low number of failures and variability in the clinical features of these patients led to low statistical significance. Interestingly, a context in which our molecule performed worse was the LVAD infection, for which no data are available in the literature, and in which classic long-term suppressive strategies or bridge therapies remain the treatment of choice; nevertheless, there is not an easy explanation for this phenomenon. This topic deserves a dedicated investigation in the future. On the other hand, no failures were observed for intravascular infections. The most satisfactory results from the off-label use of DBV were derived from the OA setting, with an overall 83% rate (45/54 patients), confirming the excellent suitability of DBV in this context, as already suggested in previous works [21,22,23,24,25,26,27]. In particular, no patient failed when treated for spondylodiscitis, and the percentages of failure were low in other OA indications. From a PK/PD perspective, the C_max_ measured at the end of the first infusion appeared to be the best PK predictor of clinical success, showing similar performance with AUC parameters, particularly in patients treated for consolidation purposes. The C_max_ was, in turn, found to be significantly different between genders and negatively correlated with BSA. Taken together, these results suggest that the therapeutic schedule for DBV could be optimised based on gender differences and anthropometric features. Specifically, the dual-dose one-week-apart schedule (*group 1*) showed both a higher proportion of treatment success, as well as a higher C_max_ value at the second administration and significantly higher cumulative AUC in the first month. In terms of the capability to predict treatment failure/success, we observed significant differences in terms of C_max_ between patients who failed in treatment for consolidation purposes (partial response to previous treatments) and those who were cured: In this context, it was possible to identify a putative cutoff value for C_max_ at the first dose of 313 mg/L, which could be predicted with good sensitivity and specificity. It is noteworthy that failures in this context happened at lower concentrations than for other subgroups (e.g., in patients with previous failures to other treatments), highlighting the cut-off value of 313 mg/L as a general threshold level for risk of failures, also in other contexts. These results provide useful information for a possible “fast-track” TDM use since the significant correlation between C_max_ and outcome suggests the possibility of using it as a proxy for early PK exposure and for the prediction of success/failure [33]. On the other hand, some previously proposed candidate markers for TDM use, such as DBV concentration after 1 month of treatment, seemed to be unrelated to the outcomes in our cohort [8,9]. Together, these observations suggest that, despite its long-acting activity, the clinical effectiveness of DBV seems greatly dependent on early exposure in the first weeks of treatment, highlighting the concept of a concentration-dependent primal effect for that molecule. It may be speculated that higher concentrations and early exposure are needed in order to reach the pharmacological sanctuaries of infection, where metabolically slow bacteria can proliferate due to being protected by biofilm, creating a pabulum for resistance, infection persistence, and incomplete cure. More data from tissues (e.g., bone and bone marrow) and intracellular (PBMCs), for which collection and measurements are challenging, may clarify these points. Importantly, additional help may come from combination therapy, the use of which still remains to be evaluated in prospective comparative studies [34,35,36]. To summarise, in our cohort, failures seemed to be more ascribable to worst infection characteristics (sanctuary site, lack of previous infection control/surgery, chronic infection, and comorbidities) than to PK underexposure, except for lower C_max_ when indicated for defined purposes. In this direction, we can state that DBV is not a magic bullet but certainly a game-changing molecule in the treatment of complicated Gram-positive infections. This is of much importance, especially for comorbid patients and those with multiple treatment failures who may be exposed to unnecessary infectious risks. Other advantages presented in different works [4,5] which we could not confirm due to the nature of our study pertain to the pharmacoeconomic role of this antibiotic. Other limitations of our study pertain to the generalisability of our observations, inherent to the study design, specifically because of the lack of a prospective a priori randomised controlled schedule and the relatively low number of enrolled patients. Additionally, population modeling based on a larger population would be beneficial in predicting DBV PK parameters, and adding specific testing procedures for DBV MIC would yield additional predictive value both to the Cmax/MIC and AUC/MIC parameters. In the future, due to its microbiological and PK/PD features, it is possible to consider DBV as a first-line treatment strategy in certain indications rather than as a consolidation or step-down one [4,5,32]. 

## 5. Conclusions

In this work, we were able to show the real-life characteristics of DBV, especially PK/PD features, in terms of their correlations with anthropometric characteristics and clinical success. The high effectiveness of DBV was confirmed in a real-life cohort of treatment-experienced patients, allowing a more complete understanding of complicated OA infections. The identification of an extremely early marker as the C_max_ after the first infusion as a predictor of treatment failure could be useful for future TDM purposes. Further studies with larger and more homogeneous cohorts are needed in order to identify precise and dedicated threshold levels for DBV for each treatment indication. Finally, the observed gender-based and BSA-related differences in DBV exposure suggest the consideration of these factors for treatment optimisation.

## Figures and Tables

**Figure 1 pharmaceutics-14-01882-f001:**
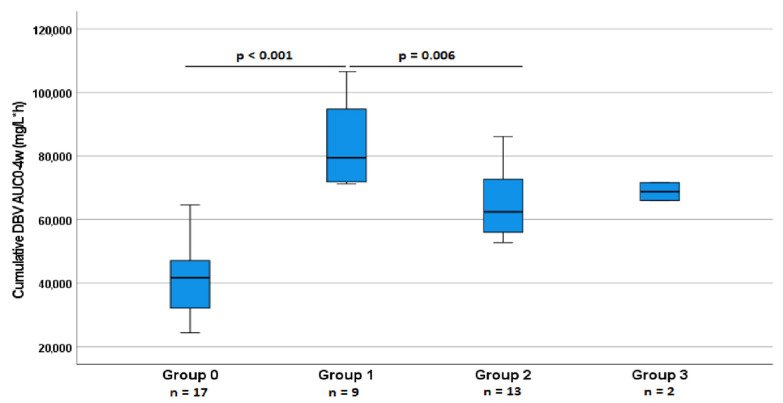
Cumulative AUC_0–4w_ for different dosing groups. DBV levels are expressed in mg/L·h. Cumulative AUC_0–4w_ of treatment was significantly higher for *group 1*, compared with all the other dosing groups.

**Figure 2 pharmaceutics-14-01882-f002:**
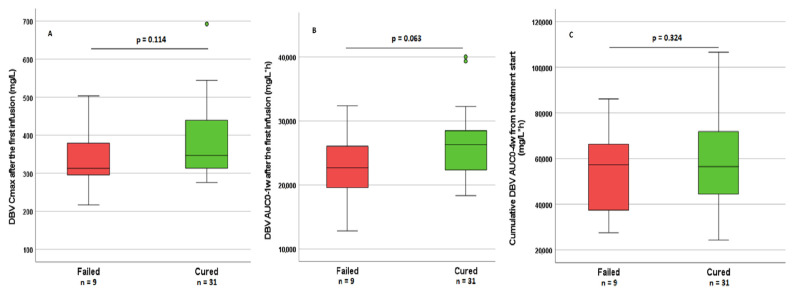
(**A**–**C**) distribution of DBV C_max_ after the first infusion, AUC_0–1w_, cumulative AUC_0–4w_, and DBV levels with regard to outcomes. Among all the PK parameters which were tested for their associations with treatment outcome, C_max_ at the first dose, AUC_0–1w_, and cumulative AUC_0–4w_ all resulted in higher values in cured patients, although these differences were not statistically significant.

**Figure 3 pharmaceutics-14-01882-f003:**
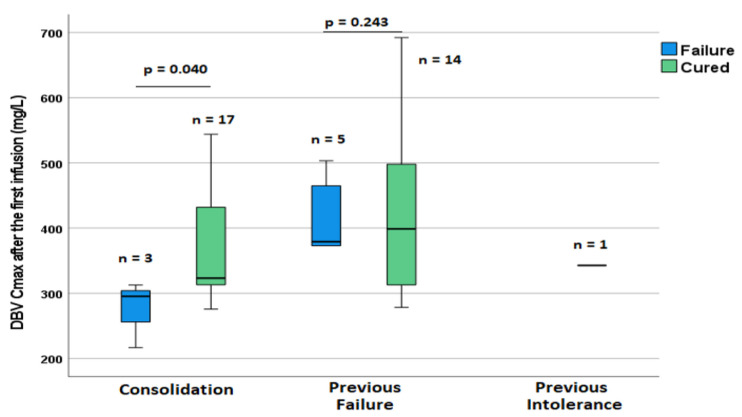
Differences in DBV levels between patient groups based on the reason for using DBV with regard to previous treatments and the outcome of therapy with DBV.

**Figure 4 pharmaceutics-14-01882-f004:**
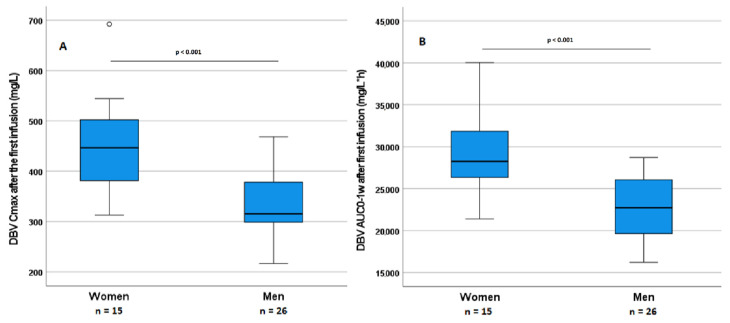
Differences in PK variables with regard to gender. (**A**,**B**) DBV C_max_ and DBV AUC_0-1w_ differences in women and men (*p* < 0.001).

**Table 1 pharmaceutics-14-01882-t001:** Overall patients’ characteristics at the baseline.

	*Overall*	*Intensive PK Study (n = 41)*			
	*n = 76*	*Group 0 (n = 17)*	*Group 1 (n = 9)*	*Group 2 (n = 13)*	*Group 3 (n = 2)*
**Variable**	Value (median [IQR]/%)	-	-	-	-
**Age (median [IQR])**	60 [50–70.5]	60 [51–73]	67 [48–75]	60 [48–71]	40 [17–40]
**Ethnicity: Caucasian (%)**	100%	-	-	-	-
**Sex: Male (n (%))**	47 [60%]	9 [53]	7 [78]	6 [60]	1 [50]
**Albumin (g/L)**	43 [40–45]	44 [39–46]	44 [43–46]	43 [42–45]	45 [40–45]
**BMI**	24.5 [22.2-28.2]	26 [22–31]	20 [22–24]	24 [23–30]	23 [21–23]
**BSA**	1.7 [1.4–2.2]	1.8 [1.5–2.3]	1.6 [1.2–1.8]	1.9 [1.7–2.2]	1.2 [1–1.3]
**eGFR (mL/min)**	94.5 [77.3–116.2]	86 [74–104]	91 [79–126]	100 [73–118]	112 [90–113]
**AEs: yes**	1	1 [6%]	-	-	-
**Infection control: yes (n, (%))**	36 [47%]	10 [60]	3 [30]	10 [80]	1 [50]
**Diabetes: yes (n, (%))**	19 [25%]	4 [23]	1 [13]	3 [23]	-
**Previous therapy: yes (n, (%))**	73 [96%]	17 [100]	8 [89]	13 [100]	2 [100]
**N. therapy lines: 1 (n, (%));** **2: (n, (%)); 3: (n, (%))**	52 [75%]; 20 [26]; 1 [1]	12 [67]; 5 [33]	4 [50]; 4 [50]	5 [40]; 8 [60]	1 [50]; 1 [50]
**Length of therapy before DBV (days)**	14 [14–35]	14 [14–56]	35 [23–75]	35 [28–105]	35 [14–35]
**Reasons for DBV (n, (%)): ** **previous failure; consolidation; intolerance**	31 [41%]; 43 [56]; 2 [3]	9 [53]; 8 [47]	5 [55]; 4 [45]	4 [30]; 7 [55]; 2 [15]	1 [50]; 1 [50]

**Table 2 pharmaceutics-14-01882-t002:** Main clinical and microbiological characteristics and outcomes.

	*Overall (76)*	*Intensive PK Study (n = 41)*			
		*Group 0 (n = 17)*	*Group 1 (n = 9)*	*Group 2 (n = 13)*	*Group 3 (n = 2)*
**Type of infection (n, (%))**	ABSSSIs: 16 [21] LVAD: 3 [4] Endocarditis: 3 [4] OAs: 54 [71] Osteomyelitis: 13 [25] Spondylodiscitis: 8 [15] Septic arthritis: 5 [11] PJI: 27 [49]	ABSSSIs: 7 LVAD: 2 Septic Arthritis: 1 Osteomyelitis: 2 Prosthetic infection: 5	Septic arthritis: 1 Osteomyelitis: 4 Spondylodiscitis: 3 Endocarditis: 1	ABSSSIs: 2 Septic arthritis: 1 Osteomyelitis: 3 Spondylodiscitis: 2 PJI: 5	Spondylodiscitis: 1 PJI: 1
**Aetiology (n, (%))**	MSSA: 32 [42] MRSA 23 [30] MRSE 7 [9.2] MSSE 2 [2.6] S. Lugdunensis 1 [1.3] Streptococcus spp.: 3 [4.1]	MRSA: 4 MSSA: 8 MRSE: 3	MRSA: 3 MSSA: 3 MRSE: 1 S. dysgalactiae: 1	MSSA: 8 MRSA: 4 MSSE: 1	MRSA: 2
**Outcome: cure (n, (%))**	58/71 [82%]	14/17 [82]	9/9 [100]	7/13 [54]	2/2 [100]
**Time to cure (days)**	28 [28–56]	14 [14–56]	84 [49–105]	84 [84–112]	28 [28–84]

**Table 3 pharmaceutics-14-01882-t003:** Main PK and PK/PD characteristics, by dosing group, including protein-binding-adjusted (PBA) parameters. -: not applicable.

		Intensive PK Study			
Variable	Overall	Group 0	Group 1	Group 2	Group 3
**C_max_ I dose (mg/L)**	342.8 [290.5–424.0]	419.6 [306.8–477.0]	347.0 [292.4–479.2]	323.3 [305.8–385.4]	383.8 [335.5–383.8]
**C_max_ II dose (mg/L)**	379.5 [321.3–448.8]		420.6 [380.6–469.8]	337.0 [290.7–424.0]	432.440 [372.2–432.4]
**AUC_0–1w_ (mg/L·h)**	-	26,693 [20,842–28,630]	25,623 [23,335–28,347]	23,924 [20,685–26,501]	28,489 [25,573–28,489]
**AUC_0–2w_ (mg/L·h)**	-	34,168 [25,336–37,301]	-	31,686 [26,281–33,914]	35,912 [32,891–35,912]
**AUC_0–3w_ (mg/L·h)**	-	39,270 [28,876–43,971]	-	-	39,985 [37,045–39,985]
**Cumul.** **AUC_0–4w_** **(mg/L·h)**	-	41,681 [32,055–48,180]	79,486 [71,849–95,210]	62,432 [55,860–74,010]	68,835 [66,005–68,835]
**PBA AUC_0–1w_/MIC**	-	14,946 [11,670–16,030]	14,341 [13,065–15,872]	13,395 [11,582–14,838]	15,951 [14,318–15,951]
**PBA AUC_0–2w_/MIC**	-	-	-	17,741 [14,715–18,989]	20,107 [18,416–20,107]
**PBA AUC_0–3w_/MIC**	-	-	-	-	22,388 [20,741–22,388]
**PBA Cumul.** **AUC_0–4w_** **/MIC (mg/L·h)**	-	23,338 [17,948–26,976]	44,505 [40,229–53,309]	34,956 [31,277–41,439]	38,541 [36,957–38,541]
**Terminal half-life**	-	508	633	413	437

## Data Availability

Raw data will be provided on request. Data are contained within the article and Appendix A.

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
