# Peer review of "Clinical Effectiveness and Pharmacokinetics of Dalbavancin in Treatment-Experienced Patients with Skin, Osteoarticular, or Vascular Infections"

_pharmaceutics, 2022, doi:10.3390/pharmaceutics14091882_

Round 1

Reviewer 1 Report

The study of PK/PD is well done, but the clinical inferences, ie cure, cannot be supported by the presented data without a controlled, blinded trial. Thus, the information presented is limited to the blood levels of antibiotic using differing dosing regimens. 

A few other comments:

In treating patients with less acute infections, measuring blood levels of antibiotics, especially those with higher protein binding properties, may not accurately reflect antibiotic efficacy; it would be tissue levels at the site of infection or intracellular that would be more relevant. In this regard, there were no tissue levels obtained.

For cure rates, it would be important to have at least a 3-6 month period of follow-up to define a cure, along with microbiological data. No such information is given here. And statements regarding optimal dosing for clinical infections are unsupportable.

Also for cure rates, a median of 56 days does not seem to favor the use of this antibiotic.

There is no need for groups 2 and 3, as there would be no clinical situation in which dosing with any antibiotic would be 3 or 4 weeks apart.

Reviewer 2 Report

The manuscript describes the results of clinical study on dalbavancin in a group of outpatients with Gram-positive infections of skin and skin-associated structures. The authors based on the data collected from a very limited number of patients recommended the optimal dosage scheme and a new marker for therapeutic drug monitoring for patients receiving dalbavancin due to osteoarticular infection. The results presented may be interesting for some clinicians, however, the quality of the manuscript is not satisfactory and several issues need clarification:

The analytical method used and validation parameters are not presented.

The authors should present pharmacokinetic profiles for each patient (or the mean values for each group of patients).

Four groups of patients 1-4 and group 0 have been identified (p. 4), whereas in Table 1 the authors  showed groups 0-3.

It is not clear when exactly blood samples were collected at weeks 1-4.

The method to assess plasma protein binding was not described.

It cannot to be concluded that Cmax is more appropriate that AUC/MIC index for TDM purposes as the authors did not determine MIC values, thus, the direct comparison of both methods was not possible.

The results of statistical analysis with such small number of patients in each group are not sufficient to provide dose recommendations for TDM. The same applies to the correlations between PK parameters and patient characteristics. It seems that population modeling approach may be more appropriate in this case.

The sentence: The clearance of DBV is not substrate …. is not clear (p.2).

Reviewer 3 Report

In this paper was described the PK of DBV. The work is well done and organized. The authors show the real-life characteristics of DBV, especially PK/PD. 

Only a request , you calculate and add the ANOVA test, comparing the means of your groups of data sets and show to what extent they differ.  This test shows better the significance of your data

Reviewer 4 Report

This is an interesting and important article discussing the off-label use of dalbavancin, however I do have a couple of suggestion and comments. 

Line 123: Did you base each dose on going through the guidelines again or did you have an internal dosing guideline that was derived from published guidance? 

Line 148: What was the justification for end of infusion and 1h sample, rather than more samples in the first 24 hours? Was the sample in the consecutive weeks taken before dose or after dose? Why was measurement of Cmax not considered? 

Line 165: What was the LOQ for dalbavancin measurement? 

Results

Supplementary material figure quite blurry, perhaps possible to make in better quality.

Line 192: immunocompromised?

Line 203: What was the reason of treatment failure? And how was failure defined?

Supplementary 2, Figure 2 are also quite blurry and could be made in better quality/more readable. 

Discussion

 Please discuss the sampling frequency as a potential limitation. Also, there have been studies that have shown dalbavancin exposure in bone long after treatment has stopped - is that a potential threat for developing drug resistance? It is great for treatment of osteoarticular infection, but might be a threat if the exposure is high many months later. 

Round 2

Reviewer 1 Report

The data presented are insufficient to support the clinical inferences of  Dalbavancin's efficacy.

Author Response

We thank Reviewer#1 for his/her comment. 

Reviewer#1 states "The data presented are insufficient to support the clinical inference of Dalbavancin's efficacy".  

We respect the work from the reviewer and we are grateful to him/her because, after the first round review, the quality of the manuscript has now improved.  However, we feel we should stress a few points to clarify our position on this point.   Importantly, we should not forget that this work has been submitted to be possibly part of the issue "Therapeutic Drug Monitoring and Pharmacokinetics-Based Individualization of Drug Therapy".  We believe Dalbavancin's efficacy is not the principal focus that has been proposed by the Editorial board for this issue and we worked accordingly, presenting data on individualization of drug therapy based on TDM and PK; at the same time, as clinical searchers, it is impossible to disregard and overlook the clinical findings in our cohort study during the observational analysis; that is why we accordingly included in the manuscript the clinical characteristics of our patients and their outcomes, highlighting the inherent limitations of this study.    In view of that, we would like to point out that the primary aim of the study is not to attest the efficacy of dalbavancin, which it has already extensively described with better designed studies for the purpose.   This is a PK and PK/PD study in real life clinical practice, with a focus on TDM and individualization of therapy.  In this work we tried to describe the PK/PD characteristics of the molecule and the differences we found between sexes and other anthropometric features (BMI, BSA); we also highlighted a possible easy to use and informative TDM threshold for identifying patients at risk of failure, and a difference in pharmacological parameters when it comes to different administration schedule (1 week apart, 2 week apart etc.). We believe these data may be useful in the individualization of drug therapy.  
By definition, our study differs from the contexts in which the PK and PK PD of dalbavancin was evaluated in the pivotal registrative studies (e.g. RCTs).  Hence, in the context of our study it is not possible to have a design with numbers decided a priori for each arm, because the study is of observational nature, on the actual use of the molecule. We clearly state that in the "limitations", at the end of the "Discussion" section. For what pertains to clinical inference from our data, we also state that the external validity of our findings needs to be verified in differently designed studies. 

Reviewer 2 Report

The Authors properly addressed my comments and suggestions. The quality of the manuscript is now significantly improved.

Author Response

We thank #Reviewer 2 for his/her comment